# Temporal Dynamics of Plasma Catecholamines, Metabolic and Immune Markers, and the Corticosterone:DHEA Ratio in Farmed Crocodiles before and after an Acute Stressor

**DOI:** 10.3390/ani14152236

**Published:** 2024-07-31

**Authors:** Andre A. Swanepoel, Christoff Truter, Francois P. Viljoen, Jan G. Myburgh, Brian H. Harvey

**Affiliations:** 1Centre of Excellence for Pharmaceutical Sciences, Department of Pharmacology, North-West University, Potchefstroom 2531, South Africa; andreswanepoel26@gmail.com (A.A.S.); francois.viljoen@nwu.ac.za (F.P.V.); 2Stellenbosch University Water Institute, Stellenbosch University, Private Bag X1, Matieland, Stellenbosch 7603, South Africa; jctruter@sun.ac.za; 3Department of Paraclinical Sciences, Faculty of Veterinary Science, University of Pretoria, Onderstepoort 0110, South Africa; jan.myburgh@up.ac.za; 4South African Medical Research Council Unit on Risk and Resilience in Mental Disorders, Department of Psychiatry and Neuroscience Institute, University of Cape Town, Cape Town 7505, South Africa; 5The Institute for Mental and Physical Health and Clinical Translation, School of Medicine, Deakin University, Geelong 3220, Australia

**Keywords:** farmed crocodiles, stress response, monoamines, heterophil-to-lymphocyte ratio (HLR), corticosterone-to-dehydroepiandrosterone (CORT:DHEA) ratio

## Abstract

**Simple Summary:**

Crocodilians have survived for millions of years and, like humans, experience stress and its subsequent consequences. Stress can cause significant economic and animal losses on crocodile farms if not monitored and managed appropriately. This study aims to identify immediate and delayed stress indicators in the Nile crocodile and to observe any release patterns after being subjected to acute restraint stress. We will consider known as well as novel stress indicators, such as dehydroepiandrosterone (DHEA), whilst observing how these novel indicators correlate with that of the known stress biomarkers. Acute restraint-stress induced noticeable changes, especially in rapid response indicators like adrenaline and noradrenaline. These monoamines, along with the stress hormone corticosterone (CORT), had a lasting impact, increasing the amount of glucose and certain immune cells in the blood. Connections were found between CORT and lactate after the acute stress reaction, whereas stress affected glucose and the heterophil-to-lymphocyte ratio (HLR) over time. Also, acute stress led to changes in the CORT:DHEA ratio, showing its quick reaction to adversity and confirming its relevance when studying the stress response in crocodilians. This research sheds light on how Nile crocodiles respond to an acute adverse event, which could improve stress management practices in crocodilians, thereby benefiting captive and wild populations.

**Abstract:**

Commercial crocodilian farms face significant economic and livestock losses attributed to stress, which may be linked to their adopted husbandry practices. The development of appropriate and modernized husbandry guidelines, particularly those focused on stress mitigation, is impeded by the limited understanding of the crocodilian stress response. Fifteen grower Nile crocodiles were subjected to simulated acute transport stress, with blood samples collected at various intervals post-stress. Plasma levels of corticosterone (CORT), dehydroepiandrosterone (DHEA), adrenaline, and noradrenaline were determined using high-performance liquid chromatography. Glucose and lactate were measured using portable meters and the heterophil-to-lymphocyte ratio (HLR) was determined via differential leucocyte counts. Significant differences were elicited after the stressor, with acute fluctuations observed in the fast-acting catecholamines (adrenaline and noradrenaline) when compared to the baseline. Downstream effects of these catecholamines and CORT appear to be associated with a persistent increase in plasma glucose and HLR. Lactate also showed acute fluctuations over time but returned to the baseline by the final measurement. DHEA, which is used in a ratio with CORT, showed fluctuations over time with an inverted release pattern to the catecholamines. The study highlights the temporal dynamics of physiological markers under acute stress, contributing to our understanding of crocodilian stress and potentially informing improved farming practices for conservation and sustainable management.

## 1. Introduction

Stress is an inherent facet of life and the responses it elicits can have both beneficial and detrimental effects on an organism’s survival. To mitigate the detrimental effects of stress, it is essential to understand the stress response at both behavioral and biological levels, especially its regulatory processes. Unlike in humans and various vertebrates, stress in crocodilians has not been thoroughly investigated from a neurological, behavioral, and psychological perspective [1].

Crocodilians are the world’s largest extant reptiles with the Nile crocodile, *Crocodylus niloticus*, being the second largest member of the Crocodylidae family [2]. Nile crocodiles in Africa faced near extinction by the mid-1970s due to their highly valued hides [3]. Their populations have, however, made a strong recovery, in part due to trade restrictions legislation [4] and the significant contribution of commercial crocodilian farms (CCFs; [3,5,6]). Indeed, CCFs have reduced the need for wild harvesting of crocodilia by supplying crocodilian products in a sustainable manner [4,6].

Crocodilians, especially farmed crocodiles, are highly stress-sensitive, which exerts detrimental effects on their welfare and the commercial value of their hides [7,8]. It is therefore crucial for CCFs to remain free of environmental stressors [2]. In fact, unexpected deaths on a CCF are often directly related to sub-optimal husbandry conditions [9]. Therefore, for CCFs to remain economically viable, sound husbandry techniques and management practices are required [2]. Unfortunately, the development of appropriate and modernized husbandry guidelines, particularly those focused on stress mitigation, is impeded by the limited understanding of the dynamics in the biological and behavioral stress axis of crocodilians.

As in mammals, crocodilians attempt to mitigate a stressor through inherent allostatic responses. Allostasis is a dynamic homeostatic process that draws on physiological and behavioral changes aimed at learned adaptation and survival [10,11]. An allostatic response differs between acute and chronic stressors. The stress response in crocodiles (see Figure 1) is essentially regulated via the sympathetic adrenomedullary (SAM) axis and the hypothalamic–pituitary–adrenal (HPA) axis [10,12] or, more specifically, in crocodiles, fish, and amphibians, the hypothalamic–pituitary–interrenal (HPI) axis. The acute response is mediated by the SAM axis and is dedicated to immediate survival requirements while neglecting energy storage or reproduction [10,13]. On the other hand, extended responses involve the HPA/HPI axis and elevation of glucocorticoids. The latter results in prolonged reproductive and immune suppression, as well as muscle atrophy and maladaptive behaviors when chronically elevated or when released in an unregulated fashion. If these effects persist without intervention, they collectively contribute to allostatic load [11,14]. Importantly, corticosterone (CORT) provides negative feedback on the HPI-axis at the hypothalamus and pituitary to prevent excessive release of CORT [15,16].

In crocodilians, the SAM axis involves the release of adrenaline and noradrenaline from the adrenal glands [10,16], which induces hyperglycemia, increased blood flow to essential limbs and organs, and increased heart rate and respiration [17,18]. The HPI axis is activated by physiological, environmental, or physical stressors [13,19], resulting in the sequential release of corticotropin-releasing hormone (CRH), adrenocorticotropic hormone (ACTH) [4,19], and CORT (see Figure 1 for detail). CORT, the principal stress hormone in crocodilians [18,19,20], induces glycolysis, gluconeogenesis, glycogenolysis, and proteolysis, to mobilize energy substrates [21,22], leading to increased blood glucose and lactate. Lactate and lactic acidosis are produced during strenuous exercise and are potentially harmful [23]. Dehydroepiandrosterone (DHEA) is a metabolic intermediate in gonadal steroidogenesis [19,20,24] that exerts anti-stress effects such as improved immunological functioning, and reduced anxiety [20,24]. CORT decreases inflammatory mediators and decreases specific immune cells [25], e.g., lymphocytes, basophils, eosinophils, monocytes, and macrophages, while increasing heterophils (analogous to mammalian neutrophils) [26]. The heterophil-to-lymphocyte ratio (HLR) highlights the effects of stress on the immune system, specifically the time-dependent downstream effects of CORT [10,27].

Allostatic load presents with failure to reproduce and to thrive, anxiety or agitation, and an increased susceptibility to infection [28]. While the immediate effects of SAM axis activation are evident, the enduring effects of the HPI axis activation ensure long-term homeostasis and recovery but, at the same time, can be severely disabling to the animal if left unchecked.

Lance and Elsey [7] investigated the effects of restraint stress and were the first to report the acute and prolonged reaction of catecholamines in American alligators (*Alligator mississippiensis*) and their association with CORT, glucose, and white blood cells. They found that CORT levels rose progressively for the first 4 h, declined at the 8- and 24-h collection point, and then rose again at 48 h, whereas glucose rose significantly by 24 h and remained elevated for 48 h. White blood cells, including HLR, showed fluctuations indicative of immune system suppression. They concluded that handling alligators, taking multiple blood samples, and keeping them restrained for more than 8 h is a severe stressor and therefore an important component of stress studies in this species.

Similarly, Franklin et al. [29] and Pfitzer [18] studied the effects of different handling stressors on CORT, glucose, lactate, and various hematological markers in estuarine (*Crocodylus porosus*) and Nile crocodiles, respectively. Franklin et al. [29] measured the hematocrit and hemoglobin concentration as well as the plasma CORT, glucose, and lactate levels. While electrical stunning and physical restraint was stressful, the magnitude of the stress response and post-stress recovery times were significantly less in the electrically immobilized/stunned animals versus physical restraint, emphasizing the suitability and translational validity of the latter for emulating capture/relocation stress. The results from Pfitzer [18] coincided with those of Franklin et al. [29]. Furthermore, Pfitzer [18] analyzed novel blood enzymes as potential stress biomarkers, such as alanine aminotransferase (ALT), aspartate aminotransferase (AST), alkaline phosphatase (ALP), and creatinine kinase (CK), although their results proved inconclusive.

More recently, Duncan et al. [30] compared CORT, glucose, lactate, and calcium levels in *Melanosuchus niger* and *Caiman crocodilus*, identifying differences in the lactate clearance rates as an important aspect of crocodilian post-capture recovery [23]. Their results showed that different crocodilian species exhibit diverse HPA axis activations to stressors and that some species may be more susceptible to acute stress than others.

Finger [31] focused on establishing CORT and immune function levels in hatchling estuarine crocodiles using bacterial killing assays (BKA). CORT levels were found to be lower than that reported previously in saltwater crocodile hatchlings and, surprisingly, were not significantly associated with growth (head length) or immune parameters. Importantly, crocodiles did not experience stress under their housing conditions, which adhere to the Australian industry Code of Practice, a guideline also used by many South African crocodile conservation facilities (CCFs). Additionally, the study underscored the importance of measuring CORT and immune biomarkers to assess stress and husbandry conditions in crocodiles. Lastly, Lance and Elsey [32] studied the interactions between stress and the reproductive system in a population of American alligators of varying sizes (740 to 2120 mm total length). They found that short-term capture stress and increased CORT significantly reduced testosterone levels in male alligators.

The above-described biomarkers are linked to an array of behavioral manifestations that at first are adaptive and gradually become maladaptive [8,33]. In mammals and humans, the interaction between glucocorticoids, monoamines, and other stress-related neuropeptides is linked to various neuropsychiatric and anxiety manifestations [33,34]. Likewise, captive crocodilians encounter acute or chronic environmental, anthropogenic, and biological stressors that can result in maladaptive behaviors, culminating in diminished animal welfare and substantial economic losses for farmers [35,36]. This phenomenon presents excessive lithophagy (ingestion of stones), anorexia, hydrophobia, runting, and piling [8,37].

A deeper understanding of the crocodilian stress response is necessary, especially by assessing an array of stress biomarkers and their response over time post-stress. By examining standard acute and chronic stress biomarkers, such as monoamines, metabolic and immunological markers, and glucocorticoids in conjunction with a novel biomarker, i.e., DHEA, we will attempt to elucidate new connections within the stress response and thus provide a deeper understanding of this physiological process and how it informs on the presence of adaptive versus maladaptive behavior. This will enable future researchers, conservationists, farmers, and animal welfare specialists to identify novel and effective ways to mitigate the damaging effects of uncontrolled stress in this species.

## 2. Materials and Methods

### 2.1. Study Animals, Husbandry, and Study Layout

Fifteen “grower”/juvenile male Nile crocodiles of farmed origin and with an average total length (TL) of 1464 mm were bred, hatched, and raised on Inyoni Estate Crocodile Farm (Silkaatsnek, Brits, South Africa) where this investigation took place. The study subjects were separated from a larger group of 500 individuals using gender and TL as screening parameters. All crocodile pen specifications, as well as husbandry and welfare monitoring procedures, were based on the South African National Standards guidelines on crocodiles in captivity (SANS 631:2009) [38], with additional/improved procedures adopted from the Natural Resource Management Ministerial Council [39].

On Day 1, the crocodiles were allowed to habituate to their new pen with adjusted stocking densities for four weeks before the initial blood collection (baseline or BL). Following the acclimation period, the individuals were electrically immobilized/stunned via a custom-built electric stunner (±170 volts), microchipped (Destron LifeChip^®^, Destron LifeChip, Dallas, TX, USA; sourced from Allfex^®^, Cape Town, Western Cape, South Africa), and their TL and weight noted, whereafter blood samples were collected. The animals were then returned to their respective pens.

Feeding and general husbandry practices continued for approximately four weeks until “Day 32” when the stressors were introduced (standard housing period). The stressors included typical capture practices, viz. electrical stunning, taping shut the jaws, handling, and a simulated transport scenario where individuals were confined in hessian jute bags for 7 h. Blood collection took place at varying intervals, viz., immediately after the initial stressors (T0H), then 2 h (T2H) and 7 h (T7H) post-stressor. The crocodiles were returned to their respective jute bags after each collection. The crocodiles were returned to their experimental holding pens after the last blood collection (T7H). The study utilized animals as their own control versus that over time, enhancing baseline integrity.

The three groups were then closely monitored, including a welfare-based recording of feeding habits, for 15 days or until their feeding behavior normalized. An additional 23 days (Day 47 to Day 70), documented as a “seasonal grace period”, was incorporated to lessen the stress and its effects when the animals were relocated to their original communal pen.

All other chemicals and reference standards were purchased from Merck Life Science (Pty) Ltd. (Johannesburg, South Africa).

### 2.2. Plasma Collection and Analysis

Blood samples were collected from each crocodile by a registered veterinarian [40]. In short, blood was collected by inserting a 38 mm 20-gauge short bevel hypodermic needle into the post-occipital spinal venous sinus, where approximately 2 mL of blood was withdrawn using a 2 mL syringe. The blood was then transferred to 4 mL K2-EDTA-containing blood collection tubes. A few drops of blood were left in the syringe to measure the blood glucose and lactate concentrations and to make blood smears for the differential leukocyte counts. The blood collection tubes were then sealed and centrifuged at ±1500 rcf (±3500 rpm) for 10 min to separate the blood plasma from other blood components. After centrifugation, the plasma was transferred to 2 mL cryovials where they were placed on ice for transport. All plasma samples were flash-frozen after transportation, i.e., within 8 h of collection, using liquid nitrogen and stored in a −80 °C freezer until the day of analysis (within 21-days of collection).

The plasma sample preparation for monoamine analysis was adapted from Yang and Gouaux [41]. Monoamines were analyzed using a high-performance liquid chromatography (HPLC) system connected to an electrochemical detector (HPLC-ECD), as described by Viljoen et al. [42]. Briefly, 10 mL of Solution A was prepared, comprising 7 M perchloric acid made up to volume with 100% methanol, together with 1 µg/mL 5-hydroxy-Nω-methyltryptamine (5-HMT) added as an internal standard after which it was vortex-mixed. To a 1.5 mL microcentrifuge tube, 200 µL plasma and 5.5 µL of Solution A were added, followed by vortex-mixing for 1 min. Tubes were then centrifuged at 20,817 rcf (14,000 rpm) for 30 min at 4 °C. After centrifugation, each sample supernatant was separately transferred to autosampler amber vials to be injected into the HPLC column. The analytical instrumentation included a Dionex™ Ultimate 3000 HPLC system (Thermo Fisher Scientific, Waltham, MA, USA), having an isocratic pump and autosampler, coupled to an RS Electrochemical detector with an Ultra 6011RS Analytical coulometric flow cell (Thermo Fisher Scientific, Waltham, MA, USA). The bioanalysis was carried out using a Venusil ASB C8 HPLC column (4.6 mm × 250 mm, particle size 5 µm, pore size of 150 Å, and surface area of 200 m^2^/g) (Stargate Scientific, Roodepoort, South Africa).

The bioanalysis of plasma CORT and DHEA was performed on an HPLC system connected to a charged aerosol detector (HPLC-CAD). Solution B was prepared prior to the preparation of the plasma samples, comprising 2.5 µg/mL dexamethasone internal standard solution in distilled water that in turn contained 2% formic acid. Initially, 650 µL plasma, 100 µL of Solution B, and 3500 µL of 100% methyl tert-butyl ether (MTBE) were added to a 15 mL plastic centrifuge tube, whereafter the centrifuge tube was sealed and vortex-mixed for 2.5 min. The samples were then attached to a rotating mixer and mixed for 30 min. The samples were vortex-mixed for another minute before being centrifuged for 15 min at 4 °C and 1300 rcf (3500 rpm). After centrifugation, the clear supernatant was transferred to a 15 mL glass centrifuge tube; the liquid line was marked as a reference for later. The supernatant was then evaporated under a steady stream of nitrogen gas until the inside glass tube was dry. The residue was then reconstituted with 150 µL of 20% methanol, followed by vortex-mixing for 1 min whilst ensuring the liquid reached the demarcated liquid line. The glass centrifuge tube containing the reconstituted plasma sample was centrifuged for 10 min at 4 °C and 1300 rcf (3500 rpm). Finally, the clear supernatant from each sample was transferred into glass inserts of amber autosampler vials to be injected into the HPLC column. The HPLC instrumentation assembly included a Thermo Scientific™ Vanquish™ HPLC equipped with a Vanquish Charged Aerosol Detector F with Concentric Flow Nebulizer (Thermo Fisher Scientific, Waltham, MA, USA), connected to a Venusil MP C18 HPLC column (2.1 mm × 150 mm, particle size 3 µm; sourced from Bonna-Agela Technologies, Torrance, CA, USA).

The software management system operated for the HPLC analysis was the Dionex™ Chromeleon™ 7.2 SR5 (monoamines) and 7.3 (hormones) Chromatography Data Systems from Thermo Fisher Scientific (Waltham, MA, USA).

Blood glucose and lactate concentrations were measured via portable analytical meters as described in Pfitzer [18]. The measurement of these biomarkers was then recorded together with the time of blood withdrawal. Blood glucose concentrations were measured directly after blood collection using two Contour^®^ TS blood glucose meters and appropriate Contour^®^ TS test strips (Ascensia Diabetes Care Holdings AG, Sandton, South Africa). Blood lactate concentrations were measured using lactate strips in an Accutrend^®^ Plus System (Cobas^®^, Roche Diagnostics, Rotkreuz, Switzerland).

HLR was determined by differential leucocyte counts of fixed and stained blood smears. All blood smears for microscopical examination were prepared in duplicate using the wedge technique, according to Carr [43]. After preparation of the blood slide, it was left to dry under ambient temperatures, whereafter it was stored for later staining and examination. The slides were stained with a standard Hemacolor^®^ Rapid staining kit (Merck^®^, Johannesburg, South Africa), with differential leukocyte counts performed according to Koepke [43]. In this way, 100 identifiable leukocytes were counted per slide using a trinocular light microscope (OPTIKA ^®^ B-190 series: Ponteranica, Italy) at 60 times magnification through a 10 times ocular lens (total magnification of 600× *g*). Each smear was counted twice.

### 2.3. Statistical Analysis

RStudio Version 0.353 was used for all statistical analysis. Using the lme4 package, restricted maximum likelihood linear mixed-effect models were employed to analyze changes over time in the measured biomarkers and ratios [44]. Time was considered a fixed factor, while subjects were treated as a random effect in these models. After model fitting, post-hoc tests were conducted using the Tukey adjustment for all pairwise comparisons. Thereafter, Cohen’s *d* values were computed with a 95% confidence interval (95CI), according to Bakdash and Marusich [45] and Lenth et al. [46]. Only effect sizes classified as large (*d* ≥ 0.8) were considered significant and reported, as per Percie du Sert et al. [47]. Following the initial analyses, repeated measures correlation analyses (with type III analysis of variance; ANOVA) were performed across the various biomarkers using the rmcorr package as described by Bakdash and Marusich [45]. No outliers were excluded from the datasets due to the limited sample size (n = 15). The criterion for statistical significance was set at *p* < 0.05. When analytes were below the limit of detection/quantification, the lowest value detected for that specific analyte, divided by two, was used as described in Beal [48] and Giskeødegård and Lydersen [49]. All graphs were generated and refined utilizing Graphpad Prism^®^ (version 8; GraphPad Software division of Dotmatics, Boston, MA, USA).

Power analysis was performed using G*Power software (version 3; Universität Kiel, Kiel, Germany). The use of 15 animals was established following a sensitivity analysis (ANOVA; repeated measures, within factors) and supported by the reports of Lance and Elsey [7], Shilton et al. [37], and Du Plooy et al. [50]. An A priori analysis, set at a medium estimated effect size (F = 0.25), α error (0.05), and 80%; power non-sphericity correction of 1; and correlation among repeated measures of 0.5, confirmed this.

## 3. Results

### 3.1. Noradrenaline and Adrenaline

The initial ANOVA revealed a statistically significant effect of time for plasma noradrenaline (Figure 2: *F*_3, 42_ = 3.77, *p* = 0.018) and adrenaline (Figure 2: *F*_3, 56_ = 4.56, *p* = 0.006) concentrations. Although the BL and T0H concentrations were comparable (10.77 ± 21.15 ng/mL vs. 23.82 ± 13.85 ng/mL, *p* = 0.10, *d* = 0.86 [0.11; 1.61]), noradrenaline concentration at T2H (6.83 ± 7.27 ng/mL) was significantly decreased, compared to that at T0H (*p* = 0.02, *d* = 1.11 [0.35; 1.88]). A statistically significant difference was identified for plasma adrenaline between BL and T0H (5.70 ± 4.01 ng/mL vs. 24.93 ± 22.49 ng/mL, *p* = 0.01, 1.19 [0.43; 1.96]), whereafter it decreased toward T2H (6.76 ± 9.71 ng/mL, *p* = 0.02, *d* = 1.13 [0.37; 1.89]).

### 3.2. Corticosterone (CORT) and Dehydroepiandrosterone (DHEA)

The initial ANOVA on plasma data revealed a statistically significant effect of time for plasma CORT (Figure 3: F_3, 56_ = 10.52, *p* < 0.0001) and DHEA (Figure 3: F_3, 56_ = 6.78, *p* = 0.0006) concentrations. Although BL CORT concentrations were comparable to T0H (5.25 ± 5.56 ng/mL vs. 4.23 ± 8.99 ng/mL, *p* = 0.99, *d* = 0.10 [−0.63; 0.83]), DHEA concentrations were significantly increased between BL and T0H (5.81 ± 6.55 ng/mL vs. 14.08 ± 6.04 ng/mL, *p* = 0.002, *d* = 1.43 [0.65; 2.21]).

Plasma CORT concentrations at T2H (22.73 ± 13.26 ng/mL) were also significantly higher versus BL (5.25 ± 5.56 ng/mL, *p* = 0.0002, *d* = 1.68 [0.88; 2.48]), T0H (4.23 ± 8.99 ng/mL, *p* = 0.0001, *d* = 1.77 [0.97; 2.58]), and T7H (6.68 ± 12.13 ng/mL, *p* = 0.0007, *d* = 1.54 [0.75; 2.33]).

Plasma DHEA concentrations were significantly increased at BL versus T2H (5.81 ± 6.55 ng/mL vs. 12.76 ± 4.95 ng/mL, *p* = 0.01, *d* = 1.20 [0.43; 1.97]) and significantly lower at T0H and T7H versus T2H (14.08 ± 6.04 ng/mL vs. 8.05 ± 5.50 ng/mL, *p* = 0.002, *d* = 1.43 [0.65; 2.21]).

### 3.3. Corticosterone-to-DHEA (CORT:DHEA) Ratio

There was no influence of time on the CORT:DHEA ratio (Figure 4: F_3, 53_ = 2.573, *p* = 0.06), as indicated by the initial ANOVA, supported by the comparable values between the different time points. Interestingly, T0H values strongly tended to be lower versus BL (0.31 ± 0.61 vs. 2.97 ± 5.62, *p* = 0.07, *d* = 0.91 [0.16; 1.67]).

### 3.4. Glucose and Lactate

The ANOVA indicated a statistically significant effect of time for plasma glucose (Figure 5: F_3, 42_ = 24.51, *p* < 0.0001) and lactate (Figure 5: F_3, 42_ = 6.31, *p* = 0.001). However, the glucose concentrations at BL (3.07 ± 0.42 mmol/L) and T0H (2.65 ± 0.67) were comparable (*p* = 0.40, d = 0.58 [−0.17; 1.33]). Despite lactate levels tending to increase, similarly, said concentrations did not differ significantly between BL (1.43 ± 1.46 mmol/L) and T0H (3.20 ± 3.67 mmol/L, *p* = 0.13, d = 0.82 [0.07; 1.58]).

The average glucose concentration increased significantly from T0H to T2H (3.65 ± 0.71 mmol/L, *p* = 0.003, *d* = 1.35 [0.57; 2.14]). Furthermore, the glucose concentration at T7H (34.83 ± 1.64 mmol/L) was significantly higher versus BL (*p* < 0.0001, *d* = 2.37 [1.50; 3.24]), at T0H (*p* < 0.0001, *d* = 2.95 [2.00; 3.89]), and at T2H (*p* = 0.001, *d* = 1.60 [0.79; 2.40]).

As for lactate, concentrations increased significantly from BL to T2H (4.75 ± 2.08 mmol/L, *p* = 0.001, *d* = 1.54 [0.75; 2.34]), whereafter it returned to baseline levels at T7H (2.47 ± 2.56 mmol/L, *p* = 0.03, *d* = 1.06 [0.29; 1.82]).

### 3.5. Heterophil-to-Lymphocyte Ratio (HLR)

A statistically significant influence of time was noted on the HLR (Figure 6: F_3, 42_ = 4.33, *p* = 0.01), despite BL and T0H values being comparable (0.50 ± 0.20 vs. 0.38 ± 0.12, *p* = 0.17, *d* = 0.76 [0.01; 1.52]). Intriguingly, the HLR value at T7H (0.57 ± 0.29) was significantly higher than at T0H (*p* = 0.01, *d* = 1.25 [0.48; 2.03]).

### 3.6. Correlations

Only biomarker correlations that achieved statistical significance are presented in Table 1. Significance for Spearman’s *r* was set at varying levels of significance. A weak significance is 0.1 ≥ *r* > 0.3, a moderate significance is 0.3 ≥ *r* > 0.5, and a strong significance is *r* ≥ 0.5. This applies to all correlations with the addition of a 95% CI that should not include 0. Strong positive correlations were found between adrenaline vs. noradrenaline but also between HLR vs. glucose. Only moderate positive correlations were found between DHEA vs. CORT and Lactate vs. CORT.

## 4. Discussion

The biomarkers analyzed were found to be valid indicators of acute stress, with noradrenaline and adrenaline demonstrating rapid reactivity to a stressor, while noticeable effects on HLR were only seen over time. The CORT response displayed a range of downstream effects that translate to a varying response dependent on the biomarker assayed. Thus, a predictable strong positive correlation was observed for adrenaline and noradrenaline, as well as between HLR and glucose. Acute stress displayed temporal effects on glucose and HLR, which were strongly correlated. Finally, acute stress was associated with fluctuations in the CORT:DHEA ratio, underscoring its reactivity and swift response to a stressor and therefore validating its inclusion in this investigation.

Captive and wild animals experience different types of stressors but are dependent on the same basic adaptive mechanisms to ensure survival [48]. The initial factors involved are the SAM axis and its effector hormones noradrenaline and adrenaline. Both have short half-lives, viz, 10 to 100 s [50], and manifest change transiently. In fact, their concentrations increase within a few minutes (<2 min) following an acute environmental stressor [10] before showing a rapid decline within 60 min [19]. On the other hand, the HPI axis may take up to 15 min to show an increase in CORT post-stress, with up to an hour before many of its distal effects manifest [15,16]. This response is dependent on the duration and severity of the stressor and the integrity of the HPI negative feedback system after the dissipation of the stressor [15,16]. Catecholamines initiate immediate survival strategies, while the HPI axis hormones ensure long-term adaptation and homeostasis [10,13]. As opposed to previous studies [7,29,30], DHEA was explored both in its independent capacity and in conjunction with CORT, i.e., CORT:DHEA.

An interesting pattern emerged regarding the catecholamines (see Figure 2). While low concentrations of both noradrenaline and adrenaline were observed on Day 1 (BL pre-stress blood collection), as could be expected, baseline measurement on Day 32 that preceded the stressor (T0H) showed these values to have increased nearly 3-fold. Due to their short plasma half-life [51], these effects are unlikely to represent an adverse response to the previous four weeks of habituation/acclimatization but rather to an acute pre-emptive increase in catecholamine levels prior to the application of the stressor. Indeed, these findings concur with those of Lance and Elsey [7] who described a similar pre-stress increase in adrenaline. The underlying mechanisms are speculative but may represent some form of pre-cognition during this habituation-pre-stress period. While further study is warranted, crocodiles may experience pre-emptive stress before the handling and electrical stunning. Indeed, crocodiles emit certain vocalizations when sensing danger [8], which can induce stress in nearby groups. To mitigate this effect, only farm workers familiar to the crocodiles were present during the stunning process. Second, groups that were stunned first were recorded to ensure that there were no statistically significant differences between these groups and thus to ensure a consistent baseline was obtained. Finally, we maintained a consistent order of animals to ensure uniform recovery times between the sample collection points.

Considering noradrenaline and adrenaline (see Figure 2), both catecholamines followed a similar zig-zag pattern across the blood collection points, with a near significant trend in their absolute concentrations (*p* ≤ 0.05; *d*_NA(BL vs. T7H)_ = −0.51; *d*_Adr(BL vs. T7H_ = −0.55) at any of the collection points, reaffirming their shared release pattern. Notable differences within each analyte included the difference between T0H and T2H for both noradrenaline (*p* = 0.02, *d* = 1.11 [0.35; 1.88]) and adrenaline (*p* = 0.02, *d* = 1.13 [0.37; 1.89]), possibly indicating that T0H might have been collected near the peak of their release. There was, however, a non-significant yet notable difference between T2H and T7H (*d*_NA_ = 0.77; *d*_Adr_ = −0.48) for both monoamines. The latter is a possible secondary peak stemming from resistance to handling following the dissipation of the effects of electrical stunning.

Measured noradrenaline and adrenaline are often maligned as stand-alone stress biomarkers due to their short biological half-lives and rapid release patterns. Nevertheless, these catecholamines remain essential when investigating the acute stress response of crocodilians. When interpreted together with complementary analytes such as CORT, glucose, lactate, and the HLR, they provide a clearer picture of the stress response over time. This progression extends from its immediate “fight-or-flight” response to more complex temporal effects that determine the eventual adaptive or maladaptive capacity of the animal.

By stimulating aerobic glycolysis, gluconeogenesis, and glycogenolysis [17,52], noradrenaline and adrenaline are hyperglycemic hormones. This, together with the delayed but more sustained metabolic influences of CORT, provides the muscles with much-needed energy substrates that can be broken down and utilized during the “fight-or-flight” response [10]. The impact of catecholamines on the glycemic levels in the body therefore manifests significantly faster (“acute” effects) than that of CORT (“late” effects) [52]. This supports the pattern of plasma glucose levels depicted in Figure 5. Here, we observed a non-significant decrease in plasma glucose between BL and T0H (*d* = 0.58), suggesting the potential absence of a perceived chronic stressor and, consequently, the lack of sustained elevation in CORT levels. However, this might also be attributed to low blood glucose levels, possibly due to lower feed ingestion during the treatment period (see limitations). Studying the three collection points on Day 32, plasma glucose increased progressively and significantly with each interval/sampling point. As with other animals and humans, we expected to see an initial increase in blood glucose following the acute stressor, especially since noradrenaline and adrenaline tended to be elevated. This, however, was not the case, possibly due to differences related to the slow metabolism of crocodiles/reptiles compared to mammals [4,8]. While the acute hyperglycemic effects of noradrenaline and adrenaline may have been overlooked due to the timing of blood samples, the “late” or temporal effects of CORT, as evidenced by the gradual increase in glucose levels over time (T2H to T7H), could be responsible for this phenomenon and hence associated with the patterns of CORT activity, as discussed later. Anticipated hyperglycemia, stemming from the pre-emptive surge in catecholamines discussed earlier, was not evident in the initial blood collection (T0H), possibly due to the resulting effects on various metabolic processes, in particular, the temporal effects of CORT on glucose homeostasis [17,52].

With an increase in blood glucose forming an essential part of the acute stress response, the importance of monitoring lactate becomes evident. Hyperlactatemia is well known to be accompanied by adrenaline-induced hyperglycemia due to the release of lactate from muscle tissues for use by the liver to produce glucose [30,52]. Together with pyruvate, lactate is a by-product of glucose metabolism produced under conditions of anaerobic metabolism [29,52]. Crocodilians rely significantly on anaerobic metabolism during rigorous muscle utilization such as when captured or during naturally demanding physical activities such as hunting or territorial disputes with other crocodilians [23,53]. Even though small quantities of lactate are safe, if left unchecked, it can result in lactic acidosis and ultimately result in potentially fatal metabolic acidosis [23,54].

A strong correlation exists between lactate, glucose, and CORT [23,55], although this was only partly the case in our investigation (Figure 3 and Figure 5 and Table 1). Despite a non-significant increase in lactate (*d* = −0.72) between T0H and T2H, this trajectory coincided with a simultaneous significant increase in glucose (Figure 5) and CORT (Figure 3) during the same time interval. This increase also followed the initial increase in catecholamines (Figure 2), very likely due to increased lactate levels resulting from exhaustion following capture and restraint at T0H. Furthermore, a significant decrease in lactate was observed between T2H and T7H (*p* = 0.03). This reduction suggests an effective breakdown of lactate, thus contributing to the restoration of homeostasis. Interestingly, the moderate positive correlation of lactate with CORT (*r* = 0.40; Table 1) reaffirms the extensive interplay between different biomarkers in the stress response. Despite not having a direct relationship, lactate and CORT exhibit similar trajectories, possibly due to their shared connections with glucose, noradrenaline, and adrenaline [18].

The immune system and its subsequent inflammatory responses are especially affected by glucocorticoids [13,31]. The effect of CORT is mainly immunosuppressive and ultimately damaging if sustained [56]; it is expressed by the immunosuppressive biomarker HLR. Increased HLR suggests reduced lymphocytes and stress-induced immunosuppression [27,57] attributable to CORT-induced lymphocytopenia and heterophilia [13,26]. These actions result in a weakened inflammatory response and increased susceptibility to infection [28]. HLR is regarded as a better measure of chronic as opposed to acute stress [18], although we have observed both acute and chronic changes to the HLR in acutely stressed crocodiles herein (see Figure 6).

A slightly decreased HLR between BL and T0H (*d* = 0.761) suggests a small improvement in the acclimation status of the animals following standard housing (Day 1 to Day 32). However, the absence of a statistically significant difference does not fully support this assumption (*p* = 0.17). A significant exponential increase in HLR was nonetheless seen in the hours immediately after the acute stressor, viz. T0H vs. T7H (*p* = 0.007; d = 1.25). Importantly, these sequential and opposing findings correspond with the effects of CORT and DHEA. Between T0H and T2H we noted a nonsignificant trend toward an increase in HLR (*p* = 0.74; *d* = 0.37). While speculative, this suggests a potential association with the initial immunosuppressive effects of CORT. Thereafter, a further increase in HLR was noted between T2H and T7H (*p* = 0.09; *d* = 0.88), which resembles the temporal immunosuppressive effects of CORT. This trend coincided with the waning concentration of DHEA, potentially indicating the dissipation of its immunostimulatory effects. In support of this work, Lance and Elsey [7] also reported an acute increase in the HLR ratio following an acute stressor.

Glucocorticoid changes can be indicative of acute and chronic stress responses [10,30]. Here, CORT levels (Figure 3) increased significantly between T0H and T2H (*p* = 0.001), followed by a significant decrease at T2H versus T7H (*p* = 0.007). The absence of significant differences between BL and T0H (*p* = 0.993) could indicate the absence of chronically stressed individuals, further highlighting the gradual rise of CORT plasma levels over time. Taken together, these CORT changes not only depict the presence of a rapidly responsive HPI axis to a stressor but also one that is under tight control following the dissipation of the situational stressor, resulting in appropriate negative feedback on the HPI axis and lowering of plasma CORT levels. Interestingly, our study design and results emulate that observed in American alligators [7,29], thus providing validity for the interpretation of other co-measured biomarkers.

Importantly, the reduction in plasma CORT between T2H and T7H does not necessarily imply a diminished physiological effect. Corticosterone binding to its nuclear DNA receptor represents an immediate early gene expression event that sets in motion important metabolic, immune, and other physiological effects that are slow to begin yet become pronounced over time [22,28]. These processes cause lasting temporal effects even when the plasma concentrations are low, unlike the effects seen from catecholamines [58]. This corresponds with the sustained increases seen in glucose (Figure 5) and HLR (Figure 6) as well as the positive correlation (*r* = 0.476) between these two biomarkers (Table 1).

Glucocorticoids, like CORT, also have their limitations when viewed in isolation and are thus used with other biomarkers to bolster translational capacity [20]. That said, CORT is unable to differentiate between detrimental/prolonged versus beneficial/essential release of glucocorticoids [59] or to tease out dysfunctional states of the HPI axis [56]. DHEA is a valuable stress biomarker, although its application remains to be realized in crocodilians [56]. Being a glucocorticoid antagonist and immunostimulant, DHEA is reduced by conditions of allostatic load (see Whitham et al. [20] and Dutheil et al. [60] for review). Simultaneously, it can mitigate the effects of chronic stress. It is not surprising that it is implicated in adaptive or species-appropriate behaviors such as aggression and territorial behavior [20].

In humans, DHEA, like noradrenaline and adrenaline, is released from the adrenal glands with levels rising and diminishing rapidly [61]. Hence, it may represent an early anti-stress or stress-coping hormone. These characteristics could account for significantly increased DHEA levels observed between BL and T0H (*p* = 0.002), suggesting a pre-emptive release as discussed earlier with the catecholamines (Figure 3). Its gradual decrease between T0H and T7H (*p* = 0.03) is inversely proportional to both HLR and glucose levels over the same period. Here, the antagonistic action of DHEA on CORT weakens [20], leading to a subsequent increase in HLR (Figure 6) and glucose (Figure 5).

The moderately positive correlation between CORT and DHEA (*r* =0.32; Table 1) prompted their reinterpretation using the CORT:DHEA ratio (Figure 4), which better informs on inherent HPI axis dysfunction [20]. This ratio has been connected to immune function/immunosenescence, treatment-resistant depression, and anxiety effects in humans [20] as well as transport, environmental, and husbandry-related stressors in livestock [60,62]. As described here, the increased CORT:DHEA ratio was associated with an activation of the stress response (i.e., increasing CORT), with acutely increased DHEA levels representing the body’s attempt to reverse this imbalance, reduce stress, and restore homeostasis.

Although the CORT:DHEA ratio (Figure 4) did not show any significant interactions, except for a non-significant decrease (*p* = 0.07; *d* = 0.91) between BL and T0H, this response can represent an unabated net glucocorticoid effect. Again, regardless of the insignificance between the Day 32 collection points, the return of the ratio to near pre-stress levels (T0H) is indicative of a healthy and functional stress response and negative feedback system. Lance and Elsey [7] described a possible biphasic response to stress via CORT with a significantly higher peak emerging 48 h post-stressor, whereas Franklin et al. [29] observed this same pattern to a lesser extent at 24 h post-stressor. Neither of the latter two studies assessed DHEA or the CORT:DHEA ratio, which we believe offers utility when monitoring crocodilian welfare and/or assessing intervention strategies.

Some limitations should be noted in this study. Crocodiles are ectothermic poikilotherms and are particularly susceptible to seasonal changes. Since the study was conducted during late summer, warmer conditions might yield less variation and a more robust stress response by avoiding possible brumation effects and reduced feeding. Assessing stress-related anorexia would also have strengthened the biochemical findings and the overall conclusions. To enhance the data quality, increasing the frequency of sampling times would be beneficial, though this carries the risk of damaging the post-occipital spinal venous sinus. A potential solution is to utilize staggered groups of crocodiles.

## 5. Conclusions

The detrimental effects of stress are a major concern for CCFs, who may suffer financial and animal losses due to inadequate husbandry practices [2]. An insufficient understanding of the crocodilian stress response hampers effective stress management practices. This investigation describes an association between traditional “acute” and “late” stress biomarkers, highlighting their temporal change within the context of stress and survival. The plenitude of downstream effects of CORT is evident with simultaneous effects on lactate, glucose, and HLR, providing an indication of why its dysregulation could be deleterious to the welfare of the animal. Furthermore, concurrent use of DHEA with CORT is a useful addition to the biomarker armamentarium that may inform on the overall stress status of the animal.

## Figures and Tables

**Figure 1 animals-14-02236-f001:**
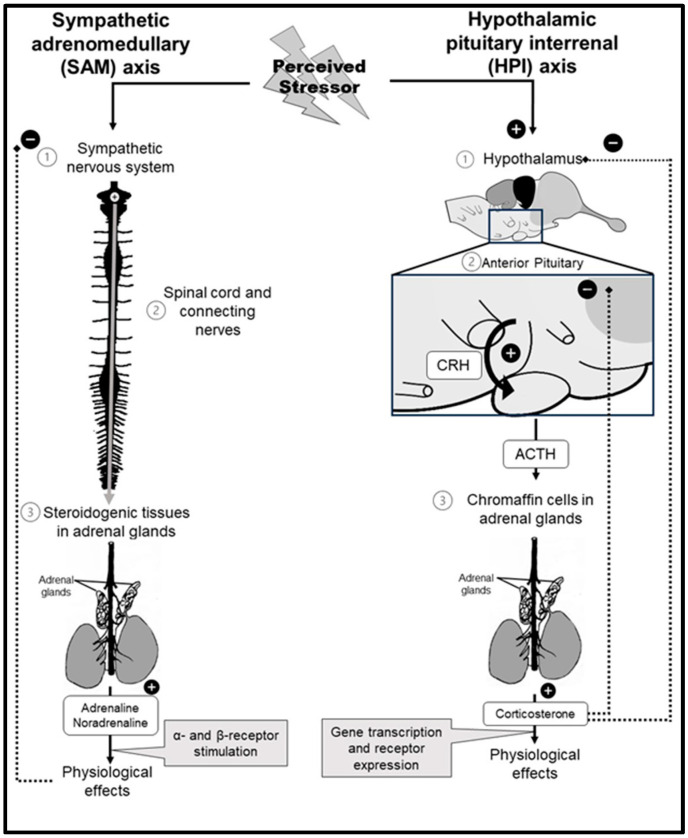
Illustration of the crocodylian HPI and SAM axes. The pathway on the left illustrates the fast-acting response of the SAM axes to a perceived stressor, wherein noradrenaline and adrenaline are released from the steroidogenic tissues in the adrenal gland. The pathway illustrated on the right illustrates the HPI axis, with CRH and ACTH regulating the release of corticosterone from the chromaffin cells in the adrenal glands. Negative feedback mechanisms of both axes are shown with the dotted lines. ACTH: Adrenocorticotropic hormone; CRH: Corticotropin-releasing hormone; HPI: Hypothalamic–pituitary–interrenal; SAM: Sympathetic adrenomedullary.

**Figure 2 animals-14-02236-f002:**
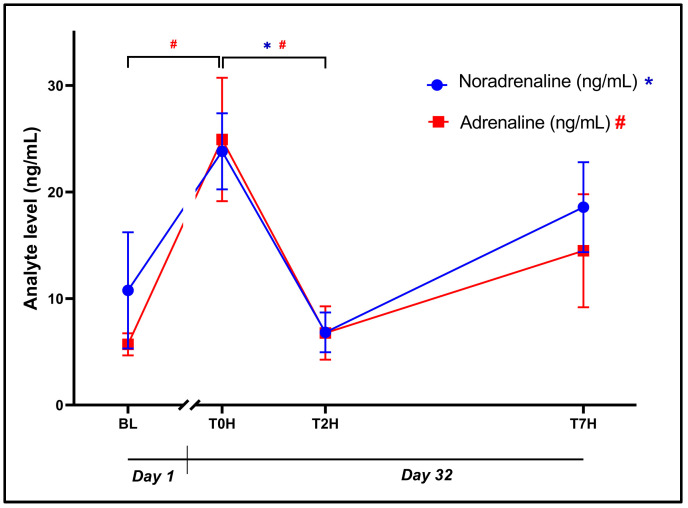
Graphical representation of the change in noradrenaline and adrenaline levels post-stress, over the time periods, as indicated, showing plasma concentrations (ng/mL) of noradrenaline (blue dots; * *p* < 0.05) and adrenaline (red squares; # *p* < 0.05 over time). Baseline blood withdrawal (BL; Day 1). Day 32 blood withdrawals with 7-h transport simulation stress: blood withdrawal immediately post-capture (T0H); blood withdrawal two hours post-capture (T2H); and blood withdrawal seven hours post-capture (T7H). Error bars indicate the standard error of the mean (SEM). Sample size (n) = 15.

**Figure 3 animals-14-02236-f003:**
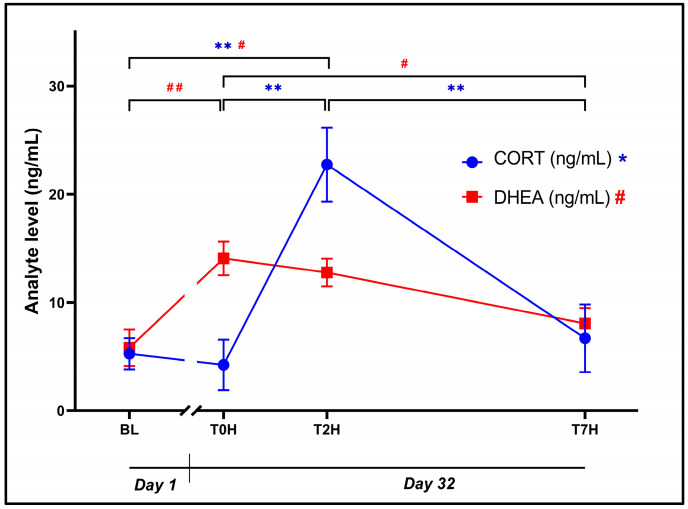
Graphical representation of the change in CORT and DHEA levels post-stress, over the time periods as indicated, showing plasma concentrations (ng/mL) of CORT (blue dots; * *p* < 0.05; ** *p* < 0.001) and DHEA (red squares; # *p* < 0.05; ## *p* < 0.001) over time. Baseline blood withdrawal (BL; Day 1). Day 32 blood withdrawals with 7-h transport simulation stress: blood withdrawal immediately post-capture (T0H); blood withdrawal two hours post-capture (T2H); and blood withdrawal seven hours post-capture (T7H). Error bars indicate the calculated SEM. n = 15.

**Figure 4 animals-14-02236-f004:**
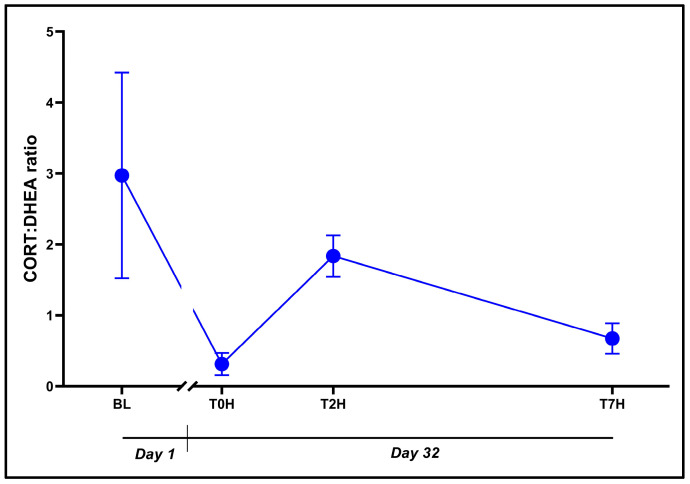
Graphical representation of the change in the CORT:DHEA ratio post stress over the time periods, as indicated. Base-line blood withdrawal (BL; Day 1). Day 32 blood withdrawals with 7-h transport simulation stress: blood withdrawal immediately post-capture (T0H); blood withdrawal two hours post-capture (T2H); and blood withdrawal seven hours post-capture (T7H). Error bars indicate the calculated SEM. n = 15.

**Figure 5 animals-14-02236-f005:**
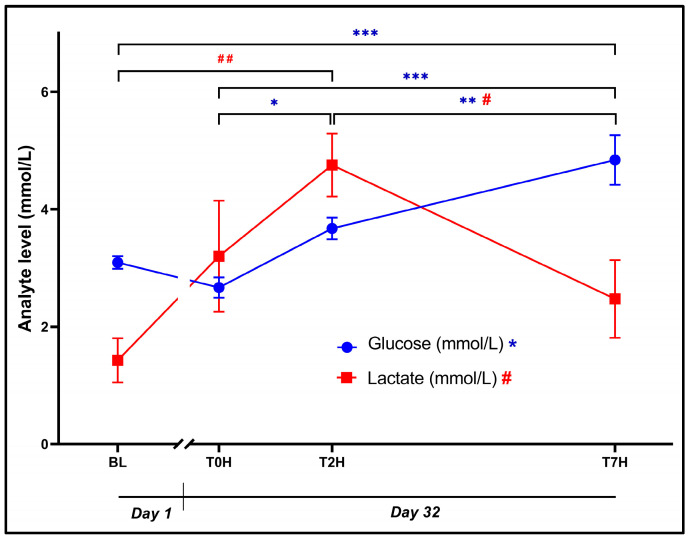
Graphical representation of the change in glucose and lactate levels post stress over the time periods, as indicated, showing plasma concentrations (mmol/L) of glucose (blue dots; * *p* < 0.05; ** *p* < 0.001; *** *p* < 0.0001) and lactate (red squares; # *p* < 0.05; ## *p* < 0.001) over time. Baseline blood withdrawal (BL; Day 1). Day 32 blood withdrawals with 7-h transport simulation stress: blood withdrawal immediately post-capture (T0H); blood withdrawal two hours post-capture (T2H); and blood withdrawal seven hours post-capture (T7H). Error bars indicate the calculated SEM. n = 15.

**Figure 6 animals-14-02236-f006:**
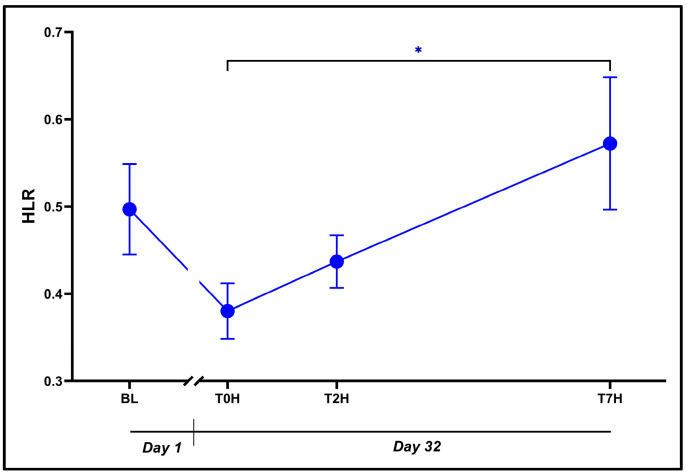
Graphical representation of the change in HLR levels (blue dots; * *p* < 0.05) post stress over the time periods, as indicated. Baseline blood withdrawal (BL; Day 1). Day 32 blood withdrawals with 7-h transport simulation stress: blood withdrawal immediately post-capture (T0H); blood withdrawal two hours post-capture (T2H); and blood withdrawal seven hours post-capture (T7H). Error bars indicate the calculated SEM. n = 15.

**Table 1 animals-14-02236-t001:** Summary of the correlations found between the various biomarkers and ratios.

Interaction	*r*-Value	Correlation	95% CI
Adrenaline vs. Noradrenaline	0.666	Strong positive	[0.519; 0.775]
DHEA ^1^ vs. CORT ^2^	0.32	Moderate positive	[0.102; 0.509]
Lactate vs. CORT ^2^	0.397	Moderate positive	[0.188; 0.571]
HLR ^3^ vs. Glucose	0.561	Strong positive	[0.384; 0.698]

^1^ DHEA: Dehydroepiandrosterone; ^2^ CORT: Corticosterone; ^3^ HLR: Heterophil-to-lymphocyte ratio.

## Data Availability

The raw data supporting the conclusions of this article will be made available by the authors upon request. Corresponding author: Professor Brian H. Harvey brian.harvey@nwu.ac.za.

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
