# Peer review of "Temporal Dynamics of Plasma Catecholamines, Metabolic and Immune Markers, and the Corticosterone:DHEA Ratio in Farmed Crocodiles before and after an Acute Stressor"

_animals, 2024, doi:10.3390/ani14152236_

Round 1

Reviewer 1 Report

Comments and Suggestions for Authors

This study, focusing on the Nile crocodile, examines physiological responses to acute negative events by simulating acute transport stress, identifying both immediate and delayed stress indicators. Through high-performance liquid chromatography, levels of corticosterone, dehydroepiandrosterone (DHEA), adrenaline, and noradrenaline in the plasma were analyzed, alongside glucose and lactate levels measured using portable meters, and changes in the heterophil-to-lymphocyte ratio (HLR). It was discovered that DHEA levels in relation to corticosterone fluctuated over time, indicating a rapid response to acute stress and underlining its importance in studying the stress responses in crocodiles. This research significantly contributes to our understanding of crocodilian stress responses, potentially guiding improved farming practices for conservation and sustainable management. However, there are some weaknesses in terms of study design, data analysis, and clarity of presentation that could be addressed.

 Concerns:

1.      Could the author clarify the specific basis for selecting the experimental subjects from the 500 individuals, and why were only male Nile crocodiles ultimately chosen? Additionally, I would like to inquire whether the authors believe that a sample size of 15 is sufficient to substantiate the conclusions of the study?

2.      Could you please address the issue of the incorrect citations, particularly the "Error! Reference source not found." mentions in line 278-279, by revising and ensuring all references are accurately cited according to the required formatting standards of the journal?

3.      In the caption of Figures, please explicitly mention the sample size in the form of "n=5" to provide clarity on the number of samples analyzed.

4.      The figure caption is missing the relevant abbreviation (T2H).

5.      What is the situation with lines 348-350? Please verify clearly!

6.      I'm unclear as to why the author does not use a consistent format for error bars in the figures?

7.      I suggest making the discussion section more concise to enhance clarity and focus, and also suggest the author engages in a discussion among reptiles and seeks similar research findings for comparative analysis.

8.      I'm curious as to why there is such a significant change in glucose concentration during the T2H stage, from 2mM to 4mM. Is such a large variation reasonable?

Comments on the Quality of English Language

None.

Author Response

Comment 1. Could the author clarify the specific basis for selecting the experimental subjects from the 500 individuals, and why were only male Nile crocodiles ultimately chosen? Additionally, I would like to inquire whether the authors believe that a sample size of 15 is sufficient to substantiate the conclusions of the study?

Response:  Thank you for this comment. The selection of male Nile crocodiles was based on the breeding practices of commercial crocodile farms, which favor males due to their growth rate and temperament. The sex of crocodiles can be influenced by the incubation temperature of the eggs, allowing farmers to produce more males if required [1]. Since the study site had a larger population of males, it was more practical to select them. The female menstrual cycle is a well-known confounder in stress studies, where female hormones can significantly affect the biobehavioral stress response. Even though the animals had not yet reached sexual maturity, we opted to focus on males to minimize this risk.

Regarding the sample size, a statistical power analysis confirmed that 15 individuals would provide sufficient data to provide robust conclusions. Moreover, other crocodilian stress studies have used similar if not a smaller number of animals
Shilton, et al. [2] and Lance and Elsey [3] and Du Plooy, et al. [4]. This we have added this in the revised manuscript on page 7, lines 296 - 302.

Comment 2: Could you please address the issue of the incorrect citations, particularly the "Error! Reference source not found." mentions in line 278-279, by revising and ensuring all references are accurately cited according to the required formatting standards of the journal?

Response:  Thank you for this comment. All references, citations and captions have now been corrected. This error likely occurred during the final formatting and copying of the manuscript. 

Comment 3: In the caption of Figures, please explicitly mention the sample size in the form of "n=5" to provide clarity on the number of samples analyzed.

Response:  Thank you for this comment. The sample size has now been added to all figure captions as "n=5", where relevant.

Comment 4: The figure caption is missing the relevant abbreviation (T2H).

Response:  Thank you for noting this. The T2H abbreviation has now been added to the captions of Figures 2 to 6.

Comment 5: What is the situation with lines 348-350? Please verify clearly.

Response: Our apologies. This error has been noted and corrected accordingly.

Comment 6: I'm unclear as to why the author does not use a consistent format for error bars in the figures?

Response: Our apologies. All graphs have now been amended to show the error bars in a consistent manner.

Comment 7: I suggest making the discussion section more concise to enhance clarity and focus, and also suggest the author engages in a discussion among reptiles and seeks similar research findings for comparative analysis.

Response: Thank you for this suggestion. We have now significantly revised the discussion section, making it shorter and more concise to improve clarity and focus. This we have noted in the revised manuscript on pages 13 to 15, lines 501 - 583. We have also revised our discussion that compares other crocodilian studies, especially those that present similar research findings. Important to note is that comparative stress studies in reptiles, especially crocodilians, is limited. We have nonetheless included relevant studies from other crocodilian species, where relevant, and have provided suitable discussion on this. At the same time, and in line with the point above, we have kept the comparative discussion concise. Caution has been exercised to avoid oversimplifying physiological similarities between different species, as well as over-extending the conclusions of the study.

Comment 8: I'm curious as to why there is such a significant change in glucose concentration during the T2H stage, from 2mM to 4mM. Is such a large variation reasonable?

Response: Thank you for this insight. This is true. However, to the best of our knowledge only two similar studies have assessed glucose levels in stressed crocodilians, viz. Lance and Elsey [3] and Pfitzer [5], who observed similar or even more significant changes in plasma glucose post-stress between baseline and the point of assessment. Their findings therefore support the observed increase noted in our study, suggesting that such changes are normal and consistent with existing literature. From a physiological point of view, we normally find changes similar to that in humans post stress. An increase in plasma glucose is caused by a combination of systems involved in the stress axis, such as glucocorticoid and monoamine release. This response is presumed to make more glucose available to the muscles and other organs that incite an appropriate “fight-or-flight” response following a stressor. This we have noted in the revised manuscript on page 12, lines 452 - 458.

References for reviews 1 and 2: 

  1. Huchzermeyer, F.W. Crocodiles: biology, husbandry and diseases; CABI Publishing: Cambridge, Massachusetts, USA, 2003; p. 352.
  2. Shilton, C.; Brown, G.; Chambers, L.; Benedict, S.; Davis, S.; Aumann, S.; Isberg, S. Pathology of runting in farmed saltwater crocodiles (Crocodylus porosus) in Australia. Veterinary Pathology 2014, 51, 1022-1034, doi:10.1177/0300985813516642.
  3. Lance, V.A.; Elsey, R.M. Plasma catecholamines and plasma corticosterone following restraint stress in juvenile alligators. Journal of Experimental Zoology 1999, 283, 559-565, doi:10.1002/(sici)1097-010x(19990501)283:6<559::aid-jez7>3.0.co;2-4.
  4. Du Plooy, K.J.; Swan, G.E.; Myburgh, J.G.; Zeiler, G.E. Electroencephalogram (EEG) assessment of brain activity before and after electrical stunning in the Nile crocodile (Crocodylus niloticus). Scientific Reports 2023, 13, 20250, doi:10.1038/s41598-023-47696-3.
  5. Pfitzer, S. Physiological parameters of farmed Nile crocodiles (Crocodylus niloticus) captured manually and by electrical immobilisation. University of Pretoria, 2013.
  6. Franklin, C.E.; Davis, B.M.; Peucker, S.; Stephenson, H.; Mayer, R.; Whittier, J.; Lever, J.; Grigg, G. Comparison of stress induced by manual restraint and immobilisation in the estuarine crocodile, Crocodylus porosus. Journal of Experimental Zoology Part A: Comparative Experimental Biology 2003, 298, 86-92, doi:10.1002/jez.a.10233.
  7. Olsson, A.; Phalen, D. Comparison of biochemical stress indicators in juvenile captive estuarine crocodiles (Crocodylus porosus) following physical restraint or chemical restraint by midazolam injection. Journal of Wildlife Diseases 2013, 49, 560-567, doi:10.7589/2012-06-160.
  8. Yang, L.; Beal, M.F. Determination of Neurotransmitter Levels in Models of Parkinson’s Disease by HPLC-ECD. In Neurodegeneration: Methods and Protocols, Manfredi, G., Kawamata, H., Eds.; Humana Press: Totowa, New Jersey, USA, 2011; pp. 401-415.
  9. Viljoen, F.; Brand, L.; Smit, E. An optimized method for the analysis of corticosterone in rat plasma by UV-HPLC: peer reviewed original article. Medical Technology SA 2012, 26, 39-42.

Reviewer 2 Report

Comments and Suggestions for Authors

Very interesting study with comprehensive hormone measurements that provides an important contribution to understanding the stress reactions of reptiles. However, there are still some unanswered questions in the material and methods section and some errors in the result section.

The introduction is well structured and reflects the current state of knowledge, which also explains well why the individual parameters are measured in this study.

There are some open questions in the material and methods section.

Line 172: Are there any data on the effects of electric immobilisation in crocodiles and reptiles? Muscle damage, skin irritation, pain, which could also affect the stress levels.

Line 194: Why do you use EDTA tubes and not serum or heparin tubes?

Line 197: Were the tubes centrifuged directly after blood collection or were they transported to a laboratory and centrifuged later?

Line 200: How long were the samples stored?

Line 202-238: Did both HPLC analyses take place at the same time or at different times and the samples were repeatedly frozen and thawed in between?

Line 254: Why was this unusual magnification chosen? The 60x only refers to the magnification of the objective, which ocular was used? Was the total leucocyte count also determined and if so, how?

In the result section and the discussion there is often listed “Error! Reference source not found” I think this is an error that happened when uploading the manuscript?

Line 277-285: The text does not match with figure 2, where there is a clear increase at T0H and a decrease at T2H.

Line 291: Please add T2H.

Line 329-332: Lactate strongly increase between BL and T0H in the figure. How can it comparable to glucose which not increase in this time?

Line 348-350. I think there is a mistake with the text?

Line 378: With regard to this study, the samples for noradrenaline and adrenaline should have been taken in a much shorter period of time, as the values were already falling again after 2 minutes, after which the samples were taken in the present study. This must be discussed. 

Line 395-401: Could this have something to do with the fact that the animals already knew that if they saw certain people, something would happen to them again and were therefore already under stress? In zoos, the animals are often stressed at the sight of the vet.

Author Response

Comment 1: Very interesting study with comprehensive hormone measurements that provides an important contribution to understanding the stress reactions of reptiles. However, there are still some unanswered questions in the material and methods section and some errors in the result section. 

Response: Thank you. We hope we have adequately addressed your comments and criticisms.

Comment 2: The introduction is well structured and reflects the current state of knowledge, which also explains well why the individual parameters are measured in this study.

Response: Thank you very much for this comment.

Comment 3: Line 172: Are there any data on the effects of electric immobilisation in crocodiles and reptiles? Muscle damage, skin irritation, pain, which could also affect the stress levels.

Response: Thank you for this comment. Given the significant concern for commercial crocodile farmers on whether electric immobilisation would negatively impact hide quality and value, such a drawback would almost certainly have been reported. While there are no explicit data on whether electric immobilisation in crocodiles can cause muscle or skin damage/irritation, or pain, Pfitzer [5] and Franklin, et al. [6] have confirm that electric immobilisation is less stressful than physical restraint methods. It also has fewer long-term adverse effects compared to chemical immobilisation [7]. Therefore, it is generally considered safer for both the animal and the handler.

Comment 4: Line 194: Why do you use EDTA tubes and not serum or heparin tubes?

Response: Thank you for this comment. The use of K2-EDTA blood tubes were used as per the original methodology articles of Yang and Beal [8] and Viljoen, et al. [9]. The K2-EDTA blood tubes allowed for the required plasma collection from the samples and since both plasma hormone analysis and plasma monoamine analysis method called for the use of K2-EDTA tubes to be used, we opted to not alter the methods. The importance of choosing only one type of collection tube was emphasized by the limited number of blood samples that could be taken safely in one day, thus the versatility of using only K2-EDTA blood tubes remained key.

Comment 5: Line 197: Were the tubes centrifuged directly after blood collection or were they transported to a laboratory and centrifuged later?

Response: Thank you for this comment. Blood collection tubes were centrifuged immediately after collection in batches of eight (the maximum capacity of the field centrifuge). The plasma was then transferred to cryovials, which were placed on ice for transportation to the laboratory. The plasma was flash-frozen using liquid nitrogen and stored at -80°C until analysis. This is noted in the revised manuscript on page 6, lines 219 - 221.

Comment 6: Line 200: How long were the samples stored?

Response: The frozen samples were analysed within 21-days of collection. This is noted in the revised manuscript on page 6, line 221

Comment 7: Line 202-238: Did both HPLC analyses take place at the same time or at different times and the samples were repeatedly frozen and thawed in between?

Response: Thank you for this comment. Samples were only defrosted once, and analysis conducted at the same time. Dr F.P. Viljoen, co-supervisor and author of the referenced article, assisted with sample preparation and analysis to ensure timely processing.

Comment 8: Line 254: Why was this unusual magnification chosen? The 60x only refers to the magnification of the objective, which ocular was used? Was the total leucocyte count also determined and if so, how?

Response: Thank you for this comment. The text has been revised to clarify the magnification used for the differential cell count. The ocular lens had a 10x magnification, which combined with the 60x objective, provided a total magnification of 600x. This magnification is sufficient for cell differentiation and counting. Total leucocyte count was not performed but could be useful in a future study more focussed on the immunological impact of stress in crocodilians. This is noted in the revised manuscript on page 7, lines 275 - 276

Comment 9: In the result section and the discussion there is often listed “Error! Reference source not found” I think this is an error that happened when uploading the manuscript?

Response: Thank you for this comment. All references, citations and captions have been corrected. This error likely occurred during the final formatting and copying of the manuscript.

Comment 10: Line 277-285: The text does not match with figure 2, where there is a clear increase at T0H and a decrease at T2H.

Response: Thank you for bringing this to our attention. This error has been corrected to indicate a decrease rather than an increase, please see the revised manuscript on page 7, line 309

Comment 11: Line 291: Please add T2H.

Response: The T2H abbreviation has been added to the captions of Figures 2 to 6.

Comment 12: Line 329-332: Lactate strongly increase between BL and T0H in the figure. How can it comparable to glucose which not increase in this time?

Response: Thank you for noting this. This section has been reworded to clarify the intended message regarding the comparison between lactate and glucose levels. This is noted in the revised manuscript on page 10, lines 355 - 365

Comment 13: Line 348-350. I think there is a mistake with the text?

Response: Thank you for bringing this to our attention. This mistake has been noted and corrected accordingly. See the revised manuscript on page 10, lines 374 - 377

Comment 14: Line 378: With regard to this study, the samples for noradrenaline and adrenaline should have been taken in a much shorter period of time, as the values were already falling again after 2 minutes, after which the samples were taken in the present study. This must be discussed.

Response: Thank you for bringing this to our attention. An improved discussion on the limitations of our sampling intervals has been added to the final paragraph of the discussion section. This is noted in the revised manuscript on page 15, lines 589 – 591. While it would be ideal to time sample collection exactly 2 minutes post-stress, practical constraints such as animal movement and the time required to stabilize and position the animal must be considered. A possible solution to this would be to use staggered groups of crocodiles, each group with their own set of withdrawal points. Each group would be subjected to the same stress and standard handling practices prior to sampling, but blood would be collected at different sets of time intervals for each group. This approach would allow for more frequent sampling at the beginning of the experiment (e.g., T0H, T0.25H, T0.5H, and T1H), providing greater clarity on the rapidly changing levels of monoamines. Subsequent groups, sampled at T2H, T4H, T6H, and T8H, or at T10H, T12H, T18H, and T24H, would help elucidate the more gradual changes in biomarkers.

Comment 15: Line 395-401: Could this have something to do with the fact that the animals already knew that if they saw certain people, something would happen to them again and were therefore already under stress? In zoos, the animals are often stressed at the sight of the vet. 

Response: Thank you for this important comment. To mitigate any pre-emptive stress, only farm workers whom the animals are familiar with, were present during the stunning procedure. These farm workers also handle daily feeding and cleaning. However, we acknowledge that crocodiles stunned last may have experienced a longer period of stress than those immobilized first. This was however, mitigated by collecting the samples in the same order for each time point to ensure consistent recovery periods. This we have noted in the revised manuscript on page 12, lines 429 - 433.

References for reviews 1 and 2:

  1. Huchzermeyer, F.W. Crocodiles: biology, husbandry and diseases; CABI Publishing: Cambridge, Massachusetts, USA, 2003; p. 352.
  2. Shilton, C.; Brown, G.; Chambers, L.; Benedict, S.; Davis, S.; Aumann, S.; Isberg, S. Pathology of runting in farmed saltwater crocodiles (Crocodylus porosus) in Australia. Veterinary Pathology 2014, 51, 1022-1034, doi:10.1177/0300985813516642.
  3. Lance, V.A.; Elsey, R.M. Plasma catecholamines and plasma corticosterone following restraint stress in juvenile alligators. Journal of Experimental Zoology 1999, 283, 559-565, doi:10.1002/(sici)1097-010x(19990501)283:6<559::aid-jez7>3.0.co;2-4.
  4. Du Plooy, K.J.; Swan, G.E.; Myburgh, J.G.; Zeiler, G.E. Electroencephalogram (EEG) assessment of brain activity before and after electrical stunning in the Nile crocodile (Crocodylus niloticus). Scientific Reports 2023, 13, 20250, doi:10.1038/s41598-023-47696-3.
  5. Pfitzer, S. Physiological parameters of farmed Nile crocodiles (Crocodylus niloticus) captured manually and by electrical immobilisation. University of Pretoria, 2013.
  6. Franklin, C.E.; Davis, B.M.; Peucker, S.; Stephenson, H.; Mayer, R.; Whittier, J.; Lever, J.; Grigg, G. Comparison of stress induced by manual restraint and immobilisation in the estuarine crocodile, Crocodylus porosus. Journal of Experimental Zoology Part A: Comparative Experimental Biology 2003, 298, 86-92, doi:10.1002/jez.a.10233.
  7. Olsson, A.; Phalen, D. Comparison of biochemical stress indicators in juvenile captive estuarine crocodiles (Crocodylus porosus) following physical restraint or chemical restraint by midazolam injection. Journal of Wildlife Diseases 2013, 49, 560-567, doi:10.7589/2012-06-160.
  8. Yang, L.; Beal, M.F. Determination of Neurotransmitter Levels in Models of Parkinson’s Disease by HPLC-ECD. In Neurodegeneration: Methods and Protocols, Manfredi, G., Kawamata, H., Eds.; Humana Press: Totowa, New Jersey, USA, 2011; pp. 401-415.
  9. Viljoen, F.; Brand, L.; Smit, E. An optimized method for the analysis of corticosterone in rat plasma by UV-HPLC: peer reviewed original article. Medical Technology SA 2012, 26, 39-42.

Round 2

Reviewer 1 Report

Comments and Suggestions for Authors

I have no further comments.

Comments on the Quality of English Language

Minor editing of English language required.

Author Response

Referee comments:

Referee #1 has made no further comments but indicates that "Minor editing of English language" is required. Unfortunately, the reviewer has not provided any context for this comment or examples where language problems occur. We have nonetheless closely scrutinized the manuscript from beginning to end and have indeed found several minor sentence construction anomalies as well as grammatical errors that we have corrected. We feel the manuscript is now much improved from an English language point of view. Thank for this.

Reviewer 2 Report

Comments and Suggestions for Authors

Many thanks for revising the manuscript. I have just a few small comments:

Lines 132 and 138 refer to reference 29, please shorten them as in line 129 to Franklin et al.

Line 216: Please add "immediately after the blood"

Line 263: Please add “directly after blood collection”

Line 358: The lactate concentration doubled between BL and T0H and the increase was not significant?

Error in line 491

Author Response

Referee comments:

  1. Lines 132 and 138 refer to reference 29, please shorten them as in line 129 to Franklin et al.
    Response: Thank you, we have now made this correction in lines 131 and 137 of the revised manuscript.

  2. Line 216: Please add "immediately after the blood".
    Response: Thank you, we have now improved this sentence, as requested. Please see the yellow highlighted text in lines 217 to 218 of the revised manuscript.

  3. Line 263: Please add “directly after blood collection”.
    Response: Thank you, we have made this correction in line 262 in the revised manuscript.

  4. Line 358: The lactate concentration doubled between BL and T0H and the increase was not significant?
    Response: We understand that the insignificant difference in concentrations between these two collection points may be surprising. The decision to analyze all bioanalysis results uniformly and to not exclude outliers was based on our limited sample size (as stated in lines 286-287). Excluding samples would have affected the overall statistical power of the data. Consequently, singular outlying samples might cause apparent differences that are not statistically significant. We believe this was the correct decision as most of our samples/biomarkers showed relatively small differences at each collection point. Thus, maintaining the statistical power of the results took precedence over singular discrepancies. Future studies and potential replications of this study will benefit from this approach, providing a reference point in a field where similar studies are rare. For the sake of transparency, we have now modified the text in the Results of the revised manuscript to indicate our awareness of this anomaly, and to bring this to the readers' attention (see lines 352-356). We also highlight this observation and our conclusion in the Discussion by referring to Figure 5 together with appropriate discussion (see line 489).

  5. Error in line 491.
    Response: This error notification has now been addressed.